

# Sexually dimorphic venom proteins in long-jawed orb-weaving spiders (*Tetragnatha*) comprise novel gene families

Pamela A. Zobel-Thropp[1], Emily A. Bulger[2], Matthew H.J. Cordes[3], Greta J. Binford[1], Rosemary G. Gillespie[4] and Michael S. Brewer[5]

[1] Department of Biology, Lewis & Clark College, Portland, OR, United States of America
[2] Division of Biological Sciences, University of California, San Diego, CA, United States of America
[3] Department of Chemistry and Biochemistry, University of Arizona, Tucson, AZ, United States of America
[4] Department of Environmental Science, Policy, and Management, University of California, Berkeley, CA, United States of America
[5] Department of Biology, East Carolina University, Greenville, NC, United States of America

Corresponding author
Michael S. Brewer,
brewermi14@ecu.edu

## ABSTRACT

Venom has been associated with the ecological success of many groups of organisms, most notably reptiles, gastropods, and arachnids. In some cases, diversification has been directly linked to tailoring of venoms for dietary specialization. Spiders in particular are known for their diverse venoms and wide range of predatory behaviors, although there is much to learn about scales of variation in venom composition and function. The current study focuses on venom characteristics in different sexes within a species of spider. We chose the genus *Tetragnatha* (Tetragnathidae) because of its unusual courtship behavior involving interlocking of the venom delivering chelicerae (i.e., the jaws), and several species in the genus are already known to have sexually dimorphic venoms. Here, we use transcriptome and proteome analyses to identify venom components that are dimorphic in *Tetragnatha versicolor*. We present cDNA sequences including unique, male-specific high molecular weight proteins that have remote, if any, detectable similarity to known venom components in spiders or other venomous lineages and have no detectable homologs in existing databases. While the function of these proteins is not known, their presence in association with the cheliceral locking mechanism during mating together with the presence of prolonged male-male mating attempts in a related, cheliceral-locking species (*Doryonychus raptor*) lacking the dimorphism suggests potential for a role in sexual communication.

# INTRODUCTION

Phenotypic differences between sexes are widespread among animals and can be attributed to sexual and/or natural selection. Sexual selection is the most common explanation for morphological and behavioral differences, which are frequently attributed to intra- or intersexual competition for mates. However, natural selection may also lead to differences

as a result of bimodal or dimorphic niche separation between the sexes (*Berns, 2013*). Recent work has highlighted the importance of additional modalities that differ between sexes, including acoustic (*Elias & Mason, 2014*) and chemical (*Wyatt, 2014b*) traits. Here we focus on sexual differences in chemical traits, and more specifically on venom, in which several studies have highlighted variation according to sex. We examine sexual differences in venom composition of spiders, and consider mechanisms of selection that may have given rise to these differences.

Venoms are complex chemical cocktails that attract research attention in both applied and basic sciences and have been characterized in many animals, including mammals (shrews, platypus), toxicoferan reptiles (lizards and snakes), fish, sea anemones, cephalopods, cone snails, insects, centipedes, scorpions, and arachnids (*Fry et al., 2009*). They can be impressively complex; among spiders in particular, individual venoms can have 1,000s of components (*Escoubas, 2006*). The complexity typically consists of related sets of molecules within which are components with exquisite functional specificity and novel activities (reviews in *Kuhn-Nentwig, Stöcklin & Nentwig, 2011*; *Smith et al., 2013*; *King, 2015*). Activities involve manipulation of physiological processes, particularly neurological. Thus they are a source for discovery of components with human applications in pharmaceuticals or insecticides (*King, 2015*).

Venoms frequently exhibit sexual dimorphism (snakes (*Menezes et al., 2006*); scorpions, (*D'Suze, Sandoval & Sevcik, 2015*; *Miller et al., 2016*); spiders, (*Herzig et al., 2008*; *Binford, Gillespie & Maddison, 2016*)). Because the primary functional roles of venoms in spiders are thought to be predation and defense, in most cases sexual differences in venom composition between adults of many species are hypothesized to result from natural selection optimizing composition based on differences in feeding biology and associated differences in diet composition and/or vulnerability to predation. Thus, hypotheses to explain sexual dimorphism in spider venoms typically center on differential optimization of chemical pools for divergent adult niches. Though largely untested, these hypotheses provide plausible explanations for many known dimorphisms. For example, in Sydney funnel-web spiders (*Atrax robustus*), male venoms are more toxic to mammals than are those of females (*Gray & Sutherland, 1978*), potentially associated with male *Atrax* being found more often wandering outside of burrows than females (*Isbister & Gray, 2004*). In contrast, in the theridiid spiders *Latrodectus mactans* and *Steatoda paykulliana*, female venoms have higher mammalian neurotoxic activity than male venoms (*Maretić, Levi & Levi, 1964*); however, as the volumes of venom were not controlled, observed differences may be due to the greater amount of venom injected by the larger females. These spiders eat vertebrates and bite humans defensively, so the general pattern of increased female potency has been attributed to the shorter lifespan of male spiders and reduced adult foraging of males (*Rash, King & Hodgson, 2000*).

The long-jawed orb-weaving spiders (Araneae: Tetragnathidae: *Tetragnatha*) provide a compelling context to explore the nature and potential cause of sexual dimorphism in venoms. Members of the genus *Tetragnatha* are broadly distributed with *ca.* 347 species worldwide (*World Spider Catalog version 19, 2018*). The majority of species worldwide are remarkably uniform in appearance, dull brown or olive in color, with long first and

second legs, typically long jaws in adulthood, and an elongate opisthosoma (*Levi, 1981*). Their behavior and ecology is also fairly homogeneous as they generally construct a light and fragile orb web with an open center and build the web over water or in other wet places (*Gillespie, 1987*), although they have undergone adaptive radiation in the Hawaiian Islands, associated with marked shifts in ecology and behavior (*Gillespie, 2004*; *Blackledge & Gillespie, 2004*).

Comparisons of crude venoms between sexes of *Tetragnatha* using 1-D protein electrophoresis have identified a particularly striking sexual dimorphism in which males have an abundance of high molecular weight components that are not present in females (*Binford, Gillespie & Maddison, 2016*). Phylogenetic comparisons across species indicate that these high molecular weight components persist across an evolutionary transition in feeding biology that reduces the differences in adult feeding niches. Specifically, adult males of orb-weaving *Tetragnatha* species do not typically build webs and are functionally wandering predators. However, a lineage of *Tetragnatha* in Hawaii has lost orb-weaving behavior (*Gillespie, 2004*; *Gillespie, 2005*), and both males and females wander in search of prey, thus reducing dimorphism in feeding biology (*Gillespie, 1991*). Males could be more prone to predation and have unique components that function in defense, but if so, increased vulnerability would also affect female wandering *Tetragnatha*. Therefore, a defensive role does not seem likely.

The goal of this study is to identify the molecules that are sexually dimorphic in venoms of a readily accessible "model" species, *Tetragnatha versicolor*. Using comparative venom gland transcriptomes and proteomes of adult males and females, we identify sequence characteristics of dimorphic components, with particular attention to those unique to males. We infer function preliminarily using homology searching and report the discovery of divergent highly expressed male-specific proteins.

## MATERIAL AND METHODS

### Collection

Individuals of *T. versicolor* were collected by hand from three populations: along the southern fork of Strawberry Creek on the UC Berkeley Campus (UCB) (37.872°N, 122.262°W), Little Sugar Creek, Binford Farm (BF), Crawfordsville, Indiana (40.061°N, 86.853°W), and Greenville, North Carolina (ECU) (35.626°N, 77.409°W).

### Venom extraction

Live specimens were transported to Lewis & Clark College where venom was extracted by electrostimulation (*Binford, 2001*). We obtained venom samples from 19 females and 16 males from UCB, nine females and two males from BF, and nine females and five males from ECU. To compile sufficient protein amounts for proteomics analyses, venom was pooled within sexes for each of these three populations.

### RNA Isolation

Trancsriptomic analyses were performed using RNA isolated only from Indiana (BF) specimens. To capture some breadth of transcriptional timing after emptying venom

glands, surviving spiders (10 females and two males) were divided into two groups within each sex, and venom glands were isolated from five females and one male two and three days after venom extraction. The glands extracted two and three days after milking were pooled within sexes and processed and analyzed separately between sexes for all subsequent analyses. To extract glands, spiders were anesthetized with $CO_2$ and venom glands were removed by dissection and flash-frozen immediately in liquid nitrogen. Total RNA was isolated by grinding tissues in TRIzol® reagent (Life Technologies, Carlsbad, CA, USA), adding chloroform (200 μL per mL of TRIzol®), mixing by inversion, and incubating for 20 min at 4 °C. The tube was centrifuged at 4 °C for 15 min at 14,000 rpm. An equal volume of cold 100% ethanol was added to the RNA-containing upper aqueous phase. The solution was then passed through an RNeasy® Mini Spin Column (Qiagen, Chatsworth, CA, USA) and purified, according to RNeasy® protocols.

## Illumina RNA sequencing and quality control

RNA extractions from the BF population of *T. versicolor* males (two) and females (10) were shipped to the Genomic Services Lab at HudsonAlpha (Huntsville, AL, USA), where cDNA libraries were prepared from total RNA (poly-A isolation, Illumina TruSeq RNA Library Prep Kit v2), and 50 bp paired-end Illumina HiSeq 2500 RNA-seq was used to generate sequence reads. All QC trimming and assemblies were done using a pipeline provided by the UC Berkeley Museum of Vertebrate Zoology (https://github.com/MVZSEQ). Quality and GC content of the resulting paired-end reads was assessed using the FastQC v0.10.0 program (http://www.bioinformatics.babraham.ac.uk/projects/fastqc/). TRIMMOMATIC v0.36 (*Bolger, Lohse & Usadel, 2014*) and CUTADAPT v1.7.1 (*Martin, 2011*) were used to clean up the sequence data: nucleotides below a quality threshold of 20 were trimmed from the ends of sequences, and sequences shorter than 36 nucleotides (after trimming) were discarded. The reads were aligned to a custom library of bacterial sequences to remove prokaryotic contamination using BOWTIE2 v2.1.0 (*Langmead & Salzberg, 2012*). Individual paired-end files were resynchronized, removing any paired-end sequences only present in one of the two files. Left and right reads that overlap were combined into a single longer read to aid in downstream assembly using FLASH v1.2.7 (*Magoč & Salzberg, 2011*).

## Transcriptome assembly and ORF prediction

The resulting male and female files were assembled separately, by sex, and combined for a general *T. versicolor* venom transcriptome. The processed reads for each sex and the combined read files were assembled using the TRINITY pipeline v2.0.6 (http://trinityrnaseq.sourceforge.net/) with default parameters except the following, group_pairs_distance = 999 and min_kmer_cov = 2. High-confidence open reading frames (ORFs) (i.e., likely coding sequences) were obtained for each gene in the transcriptome using TRANSDECODER r20140704 (*Haas et al., 2013*). A minimum protein length of 30 amino acids was used in ORF predictions. The completeness of each assembly was assessed via BUSCO v1.1 (*Simão et al., 2015*).

## Read mapping to identify transcriptome dimorphisms

The processed reads from each sex were mapped against the combined assembly to identify genes that are unique to either sex, with particular emphasis placed on male-only transcripts. The mapping was performed using STAR v2.4.2a (*Dobin et al., 2012*) and default parameters. A custom python script was used to generate a BED file from the combined transcriptome and BEDTOOLS v2.18.1 "multicov" (*Quinlan & Hall, 2010*) was used to generate counts of reads from each sex mapping to combined assembly transcripts. An additional round of mapping with BOWTIE v1.1.1 (*Langmead et al., 2009*) followed by GFOLD v1.1.4 (*Feng et al., 2012*) analysis identified differentially expressed transcripts between males and females (GFOLD cutoff = 2, which approximates a $\log_2$ fold change of two).

## Functional annotation and GO enrichment analyses

The male, female, and combined assemblies were annotated via the TRINOTATE pipeline (*Haas et al., 2013*). This approach comprises the following steps. All contigs (BLASTX) and predicted amino acid sequences (BLASTP) were searched against the Swissprot database (downloaded 23-iv-2015). Protein domains were identified by running a HMMER v3.1b2 (*Finn, Clements & Eddy, 2011*) search against the PFAM (*Bateman et al., 2004*; downloaded 23-iv-2015) database, and signal peptides, indicating secreted proteins, were discovered with SignalP v4.1 (http://www.cbs.dtu.dk/services/SignalP/). TMHMM v2.0c (*Krogh et al., 2001*) was used to annotate transmembrane domains. Finally, RNAMMER v1.2 (*Lagesen et al., 2007*) identified rRNA transcripts. Results from database searches were loaded into a sqlite database, and GO terms were applied and used in downstream analyses.

GO enrichment analyses were performed on the combined assembly using the subset of genes endemic to male spiders, as verified by the proteomic analyses outlined below. Two data sets were analyzed, (1) all male specific proteins and (2) high molecular weight (>43 kDa) male proteins. This was done using scripts provided with the TRINITY and TRINOTATE software and the R Bioconductor package "GOseq" v1.18.0 (*Young et al., 2010*).

## Gene family reconstruction via Markov clustering

Predicted ORFs from the male, female, and combined assemblies were combined into a single FASTA file, a BLAST database was created, and the sequences were searched (BLASTP; $e$-value cutoff = $e10^{-5}$) against themselves (i.e., an ALLvsALL BLAST). The results were clustered into putative gene families using the Markov Clustering Algorithm (MCL v14.137; *Enright, Van Dongen & Ouzounis, 2002*) with default parameters and an inflation value of 2.0. The resulting $e$-values were negative log transformed, and the results were separated using a heuristically chosen cutoff ($-1.91$) and visualized in CYTOSCAPE v3.0.1 (*Shannon et al., 2003*). Clusters representing putative gene families were used in subsequent analyses. Initial clustering identified a single cluster of particular interest, cluster six, comprising high molecular weight proteins that was subsequently subclustered (negative log transformed and heuristically chosen edge weight cutoff = $-0.93$).

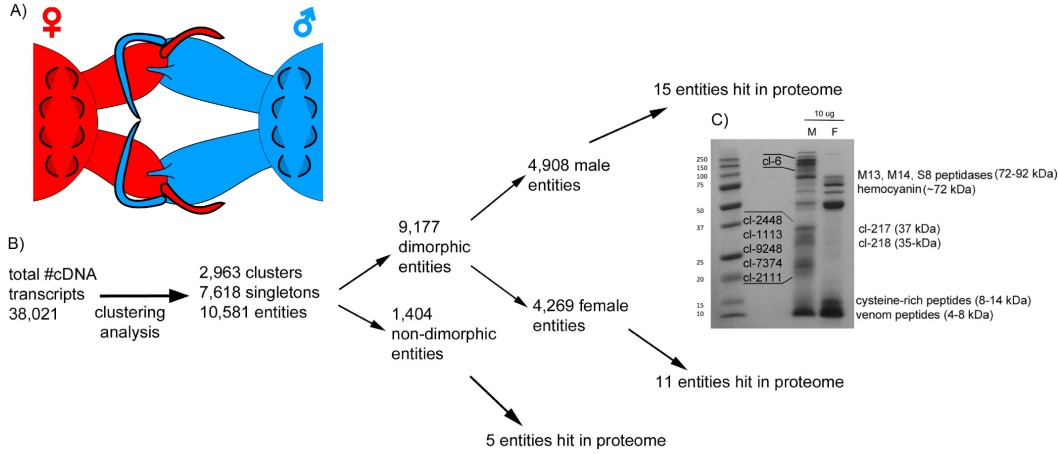

**Figure 1** **Comparative analysis of *Tetragnatha* male and female venom gland transcriptomes.** (A) Diagram shows cheliceral locking between male and female spiders during mating. (B) Flowchart shows the pipeline of transcriptome clustering analysis yielding sexually dimorphic entities detected in both transcriptomes and proteomes. (C) SDS-PAGE (12%) of *T. versicolor* crude venom from male (M) and female (F) spiders; proteins that correspond to identified clusters are labeled with "cl-" followed by the cluster number and assigned molecular weight sizes are based on predictions made from full-length amino acid sequences using the compute pI/MW tool (http://web.expasy.org/compute_pi/).

## Proteomic analyses

Crude venoms were dissolved in a standard buffer (5 mM $CaCl_2$/50 mM Tris, pH 8), pooled by sex, and shipped to the Arizona Proteomics Consortium. Venom-expressed proteins were separated by size using SDS-PAGE (12%, Fig. 1C). To increase resolution, each lane was divided into three sections and digested with trypsin followed by a clean-up step using C18 ZipTips (Millipore, Burlington, MA, USA). Tryptic peptides were analyzed using an LTQ Orbitrap Velos mass spectrometer (Thermo Fisher Scientific, Waltham, MA, USA), and the resulting MS/MS data were searched using SEAQUEST on DISCOVERER (V 1.3.0.339; Thermo Fisher Scientific, Waltham, MA, USA) against masses of theoretical fragments from a database that included our translated transcriptome sequences and all chelicerate sequences in NCBI (downloaded 4/23/2015), totaling 171,068 sequences. Matches required a fragment ion mass tolerance of 0.80 Da and a parent ion tolerance of 10.0 ppm; oxidation of methionine and carbamidomethyl of cysteine were specified in Seaquest as variable modifications. The Seaquest output was organized in Scaffold (V 4.4.3; Proteome Software Inc., Portland, OR, USA). Peptides were identified with 90% minimum threshold and 0% false discovery rate (FDR), and proteins were identified with 100% minimum threshold and 0% FDR.

## RESULTS

### Data archiving

All raw read data are available through the NCBI short read archive (SRA accession number SRP118124). Results and data files associated with proteomics (https://dx.doi.org/10.6084/m9.figshare.5378308; https://dx.doi.org/10.6084/m9.figshare.5378299), gene family

## Transcriptome

The final trimmed read files comprised 43,550,457 ♀ (20,370,960 left; 20,370,960 right; 2,808,555 merged), 48,457,662 ♂ (22,783,475 left; 22,783,475 right; 2,890,675 merged), and 92,008,100 combined (male and female reads added together) reads. The Trinity assemblies produced 16,799 ♀ (N50 = 745; 9,904,215 total bases), 24,351 ♂ (N50 = 464; 10,664,050 total bases), and 38,021 combined (N50 = 661; 20,752,384 total bases) contigs. BUSCO estimations of completeness show the combined assembly out performs the sex specific assemblies in capturing core single copy orthologs (% missing: 87 ♀; 91 ♂; 79 combined). These completeness results likely have such high percentages of missing core orthologs due to sequencing material from such highly specific venom gland tissues. Unless otherwise noted, the combined transcriptome was used in the remainder of analyses. ORF predictions produced 23,624 putative peptides, and functional annotations were obtained for 8,075 out of 28,241 Trinity identified "genes" in the combined assembly, not to be confused with contigs that comprise isoforms and alleles of individual genes.

A flow chart summary of steps we used to identify the set of transcripts ("entities") that are venom-expressed (detected in the proteome) and dimorphic is presented in Fig. 1B. We use "entities" to refer to the unique sets of venom components and define entities as the total population of clusters (homologous groups) and singleton sequences. MCL analysis identified 10,581 entities in the combined transcripts. Of these 87% are dimorphic—53% are found only in males and 47% in females. GFOLD analysis identified 3,800 (out of 23,634—16.07%) differentially expressed transcripts, indicating sexual dimorphisms in a wide range of genes including, but not limited to, venom cocktail peptides/proteins (1,430 female upregulated; 2,370 male upregulated).

## Proteome

Crude venom separations of male and female venoms show that the profiles of expressed proteins are quite different (Fig. 1C), evidence that is supported by proteomics. LC-MS/MS produced 3,205 spectra that corresponded to 62 distinct proteins in 31 clusters, which correspond to 31 entities (Fig. 1B), only 0.29 % of total entities. Of these, nine are male-specific, non-metabolic proteins; at least eight of which have no significant homology ($e \leq 10^{-5}$) with any sequences in databases searched including NCBI and Arachnoserver (Fig. 1C; Table 1A; *Herzig et al., 2010*; https:// www.ncbi.nlm.nih.gov/).

The results below detail the entities that were dimorphic and confirmed as venom components through detection in proteomes.

## Cluster 6

The most abundant, dimorphic cluster, the sixth most highly represented in the combined transcriptome (66 homologous proteins, 38,922 mapped reads), included the proteins with the highest number of distinct proteins (12) and represent 30.7 % of MS spectra in the male proteome (Table 1). Transcripts in this cluster appear to code for proteins

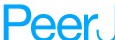

**Table 1 Sexually dimorphic components of *T. versicolor* venom.** Proteomics results are separated into four categories based on general function prediction hits from homology searches: unknown function, potentially toxic/defense proteins, potentially toxic peptides are presented in (A), and housekeeping/metabolism proteins are in (B). The symbol (†) indicates an absence of a protein in the secreted venom. The symbol (\*) indicates <1% of total #spectra. The total number of homologous polypeptides in each cluster is listed, along with the number of corresponding transcripts in the transcriptome. The symbol (ˆ) indicates a hit in the tryptic peptide search against the NCBI database.

| General function prediction based on homology searches [cluster #] | # of distinct polypeptides detected in venom | | # of spectra detected in proteome (% total) | | # of homologous polypeptides in transcriptome cluster [$e < 10-5$] (# of transcripts) | | Top hit species in homology searches (*e*-value) |
|---|---|---|---|---|---|---|---|
| | m | f | m | f | m | f | |
| **(A)** | | | | | | | |
| *Unknown function* | | | | | | | |
| New family (high MW) [6] | 12 | † | 398 (30.7) | † | 66 (38,922) | 0 (35)† | – |
| New family [1,113] | 2 | † | 25 (1.9) | † | 3 (2,639) | † | – |
| New family [9,248] | 1 | † | 10 (\*) | † | 1 (229) | † | – |
| New family [7,374] | 1 | † | 7 (\*) | † | 1 (661) | † | – |
| New family [2,111] | 1 | † | 33 (2.5) | † | 2 (1,382) | † | – |
| New family [2,448] | 1 | † | 8 (\*) | † | 2 (453) | † | – |
| New family [217] | 1 | † | 16 (1.2) | † | 7 (6,492) | 1 (195) | *Cluex* ($e10-5$) |
| New family [218] | 1 | † | 28 (2.2) | † | 8 (15,274) | 2 (3) | – |
| S8 peptidase [889] | 1 | † | 14 (1.1) | † | 1 (1,107) | 3 (886) | *Stegodyphus* ($e0.0$) |
| M14 carboxypeptidase M [116] | 1 | 1 | 6 (\*) | † | 7 (751) | 6 (766) | *Stegodyphus* ($e0.0$) |
| M13 metalloendopeptidase [11] | 5 | 1 | 501 (38.7) | 80 (4.2) | 48 (19,836) | 13 (11,358) | *Stegodyphus* ($e0.0$) |
| *Potentially toxic/defense proteins* | | | | | | | |
| Chitinase [337] | 1 | 1 | 77 (5.9) | 19 (10.1) | 6 (5,557) | 1 (2,974) | *Araneus* ($e0.0$) |
| Venom allergen 5 [843] | 1 | 1 | 34 (2.6) | 157 (8.2) | 2 (5,401) | 2 (86,981) | *Stegodyphus* ($e10-99$) |
| Hyaluronidase [10,277] | † | 1 | † | 6 (\*) | 0 (62)† | 1 (690) | *Brachypelma* ($e10-97$) |
| Phospholipase A2 (PLA2) [10,291] | † | 1 | † | 20 (1.0) | 1 (95) | 0 (73)† | *Stegodyphus* ($e10-52$) |
| *Potentially toxic peptides* | | | | | | | |
| Venom peptide [2,335] | † | 1 | † | 22 (1.2) | † | 2 (2,043) | *Nephila* BLTX631 ($e10-135$) |
| Venom peptide [846] | † | 2 | † | 270 (14.1) | 1 (139) | 3 (43,266) | *Nephila* BLTX631 ($e10-24$) |
| Venom peptide [8,293] | † | 1 | † | 187 (9.8) | 0 (40)† | 1 (13,297) | – |
| **(B)** | | | | | | | |
| *Housekeeping/metabolism proteins* | | | | | | | |
| Hemocyanin (subunits A, B, C, D, G) [19] | † | 10 | † | 835 (43.7) | 24 (6,550) | 14 (16,316) | *Stegodyphus* ($e0.0$) |
| Hemocyanin (subunit D) | † | 1 | † | 32 (1.7) | n/a | n/a | *Latrodectus*ˆ |
| Alpha amylase [4,700] | † | 1 | † | 20 (1.0) | 0 (41)† | 1 (1,777) | *Stegodyphus* ($e10-144$) |
| Alpha amylase [10,595] | † | 1 | † | 37 (1.9) | 0 (2)† | 1 (514) | *Lithobius* ($e10-98$) |
| Alpha-2 macroglobulin [453] | 1 | † | 6 (\*) | † | 3 (1,006) | 4 (997) | *Hasarius* ($e0.0$) |
| Acetylcholinesterase [831] | 1 | † | 15 (1.1) | † | 2 (9,789) | 1 (27) | *Pardosa* ($e10-159$) |
| Triacylglycerol lipase [75] | † | 1 | † | 33 (1.7) | 9 (1,636) | 6 (14,009) | *Stegodyphus* ($e10-159$) |
| Protein tyrosine phosphatase rec. [231] | 1 | 1 | 19 (1.5) | 12 (\*) | 3 (5,714) | 5 (3,671) | *Homo* ($e10-70$) |
| Corticotropin releasing factor [4,347] | 1 | † | 10 (\*) | † | 1 (138) | 1 (1,002) | *Tribolium* ($e10-82$) |
| G-protein coupled rec. (GPRmth5) [117] | 1 | † | 76 (5.9) | † | 10 (2,294) | 1 (3) | *Pediculus* ($e10-28$) |
| Beta casein | 1 | † | 6 (\*) | † | n/a | n/a | *Bos*ˆ |
| *Casein* | 1 | † | 6(\*) | † | n/a | n/a | *Bos*ˆ |
| Slit-like protein (leu-rich domain) [6,912] | † | 1 | † | 7 (\*) | 0 (2)† | 1 (100) | *Stegodyphus* ($e10-20$) |

of sizes consistent with the large proteins unique to male venoms (Fig. 1C). While none of the assembled transcripts are full length (initiating methionine through stop codon), individual transcripts in this cluster translate into proteins ranging from 41 to 1,093 aa. Multiple sequence alignment of these proteins generates a consensus sequence of 1,158 aa in length, and the longest single transcript in the alignment (1,093 aa) has a predicted MW of 128.95 kDa. The homologs in this cluster are grouped by MCL into eight sub-clusters and five singletons (Fig. 2).

Proteins in cluster 6 contain multiple repeating units, each of which has ~150 aa and a conserved pattern of 10 cysteine residues (Fig. 3). Submission of a multiple alignment of these repeats to the Fold and Function Assignment (FFAS) server (http://ffas.sanfordburnham.org/ffas-cgi/cgi/ffas.pl; PMID 15980471) returns strong evidence of distant homology to Argos, a 223-residue antagonist of epidermal growth factor receptor signaling with a known structure (PDB ID 3c9a). Argos contains three small $\beta$-sheet rich domains, the first two of which correspond to one of the sequence repeats present in the cluster 6 proteins, with a similar pattern of 10 cysteine residues making five disulfide bonds (Fig. 3B; *Klein et al., 2008*). The third Argos domain corresponds to an extra half-repeat with six of the 10 cysteines and three disulfide bonds (Fig. 3B). Argos uses contacts from multiple domains to bind and sequester small protein ligands of EGF receptors. Based on the FFAS score (−15), it is quite likely that the cluster 6 proteins have a similar fold, disulfide pattern, and domain organization to Argos; however, the sequence homology to Argos is so distant (<20% sequence identity between Argos domains 1 and 2 and any sequence repeat in the Tetragnatha proteins as shown in Fig. 3A) that a functional similarity is much less certain.

## Other male-specific venom proteins

Seven additional male-specific clusters include transcripts that code for proteins that span 20–37 kDa. These are sizes that correspond to predicted molecular weights of full-length proteins within each cluster (Fig. 1C). Cluster numbers are labeled next to individual bands in Fig. 1C and correspond to the relative rankings based on representation in the combined transcriptome: cl-2448 (>36 kDa), cl-217 (37 kDa), cl-218 (35 kDa), cl-1113 (26 kDa), cl-9248 (>26 kDa), cl-7374 (>24 kDa), and cl-2111 (21 kDa). All were detected in the proteome, though not as abundantly as cluster 6. Proteins encoded by genes in all of these clusters correspond to novel gene families without homologous sequences present in existing databases, including GenBank and Arachnoserver. Proteins in each of these clusters have multiple cysteine residues (8–21), and with the exception of clusters 217 and 218, they have at least two C × C motifs.

In addition to identifying novel families of proteins in these venoms, we also identified several sequences with evidence of homology to known enzymes. Venom proteins in males hit three large peptidases: M13 metalloendopeptidase (specifically neprilysin, ~90 kDa), M14 carboxypeptidase M (sequence not full length, but estimated to be >55 kDa), and S8 peptidase (specifically neuroendocrine convertase, ~72 kDa). Only one large peptidase—M13 metalloendopeptidase—was identified in the venom proteome of females; however, homologous sequences to each were found in both transcriptomes.
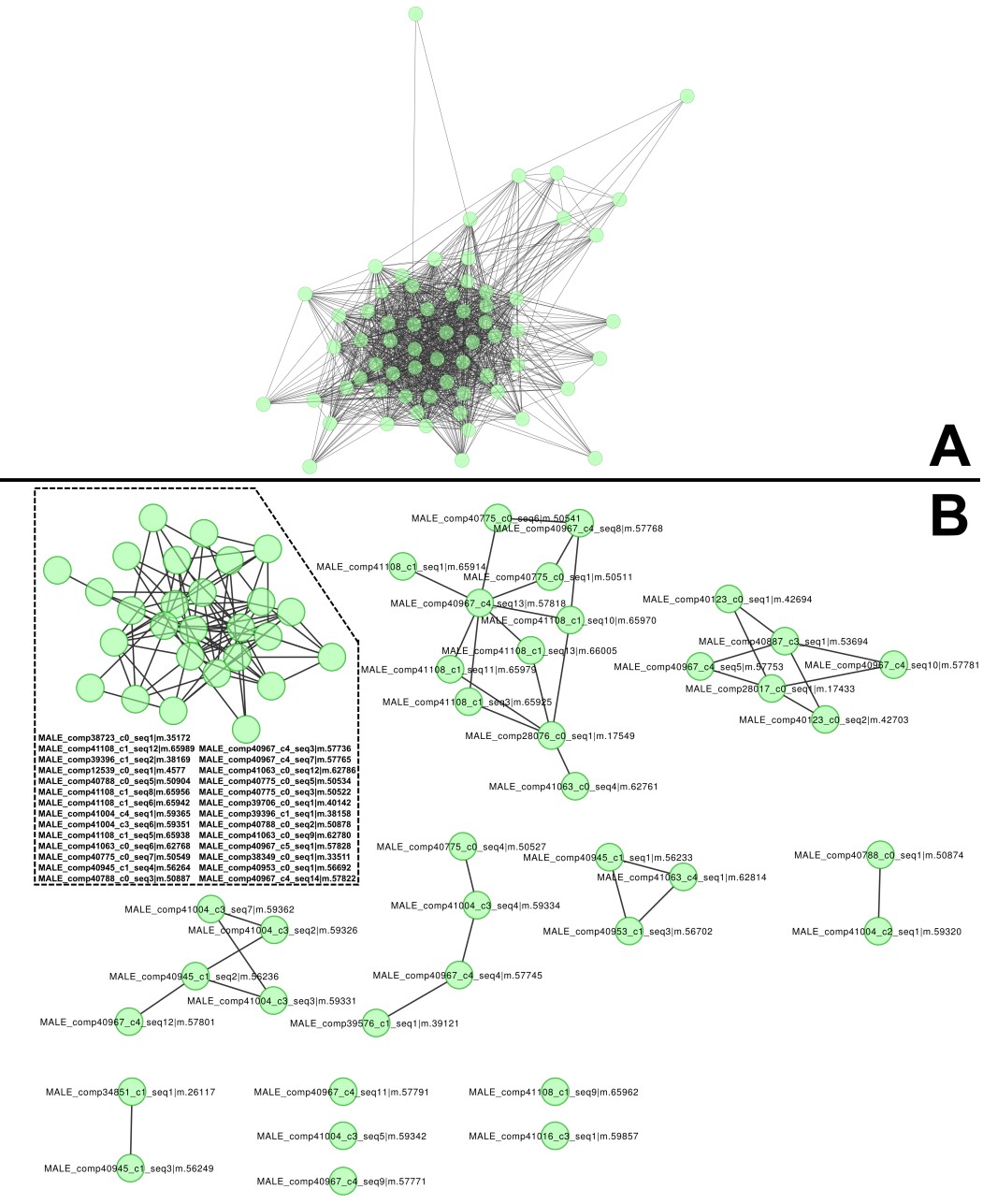

**Figure 2** **High molecular weight, male-only "gene family" and subclustering results from MCL analysis.** (A) Similarities between components of the high-molecular weight family of male-specific components, and (B) subclustering of the same family. All members are present in both networks. These components show no similarities to known venom or toxin genes, bug gene ontology (GO) enrichment tests indicate a role in hormone signaling/transport.

Both S8 and M14 peptidase family members are known to be involved with activation and processing/regulation of hormones, respectively, whereas M13 peptidases are comprised of GluZincins, a superfamily of peptidases that act on molecules <~40 aa (MEROPS peptidase database, http://merops.sanger.ac.uk/index.shtml) and have also been reported in venom

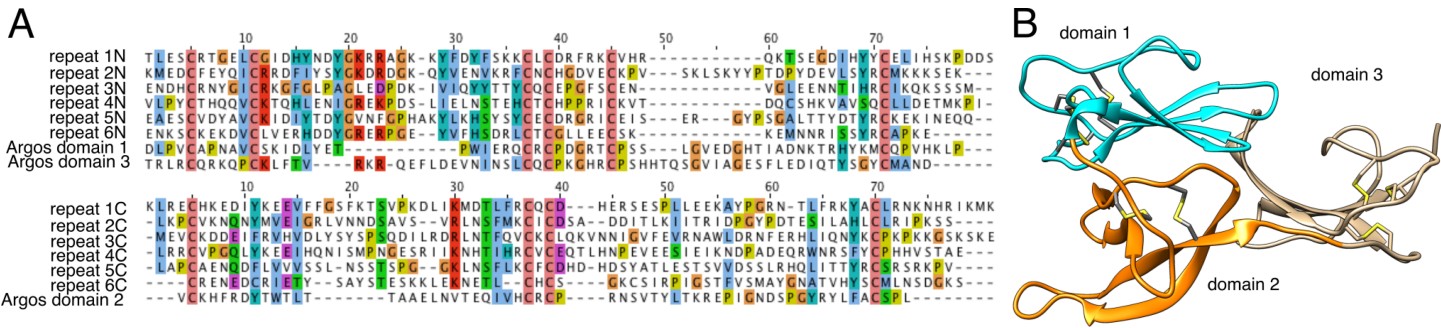

**Figure 3** Remote homology of cluster six proteins to *Drosophila* Argos. (A) Sequence alignments of Argos to each of the six repeats within a single cluster 6 protein from *Tetragnatha*; at top is the N-terminal half of each repeat aligned to domains 1 and 3 of Argos, while at bottom is the C-terminal half of each repeat aligned to domain 2 of Argos. (B) Ribbon diagram of Argos structure (PDB ID 3c9a), colored by domain with disulfide bonds shown.

of a trapdoor spider (*Undheim et al., 2013*). The degradative enzymes hyaluronidase and phospholipase A2 (PLA2) were identified in the venom of females, constituting ∼1% of the proteome. Cluster 19 contains peptides corresponding to various hemocyanin subunits and represents the most abundant set of proteins detected in the female venom proteome (Table 1B). Each subunit varies in size, and the most prominent was subunit G, which is predicted to be ∼72 kDa (Fig. 1C). The female venom is also rich in small cysteine-rich peptides, corresponding to clusters 2335, 846, and 8,293 identified in the proteome (Table 1A).

Within gene families observed only in male proteomes, we recovered interesting patterns of sexually dimorphic expression and potential functions. Despite some mRNA from females mapping to these transcripts, their peptides were not detected in the female proteome. This could be due to a lack of translation following transcription or perhaps the proteins are not present in the venom. GO enrichment analyses performed on all male-only peptides, based on the proteomics analysis, indicated potential non-feeding functions of these proteins. Four GO terms were significantly enriched, two relating to hormone functions—GO:0016486 (BP peptide hormone processing; FDR $p$-value = 0.0143), GO:0008237 (MF metallopeptidase activity; FDR $p$-value = 0.0349), GO:0042445 (BP hormone metabolic process; FDR $p$-value = 0.0349), and GO:0006518 (BP peptide metabolic process; FDR $p$-value = 0.0497). Half of the enriched GO terms were specifically associated with hormone functions, suggesting this venom-based sexual dimorphism could be involved in sexual communication.

## DISCUSSION

The results we present provide a first identification and characterization of unique and sexually dimorphic components in venoms of *T. versicolor*. Combined proteomics and transcriptomics identify proteins that are expressed in venoms and confirm the presence of sexually dimorphic expression of particular components. The majority of components we identify are sufficiently different from proteins in databases to prohibit high confidence annotation using homology searches. Recovering a low annotation percentage and high

protein-coding compliment (in terms of genes and isoforms) are both consistent with previous genomic studies of spider taxa (*Croucher et al., 2013*; *Sanggaard et al., 2014*; *Brewer et al., 2014*) and illustrate the early nature and promise of spider genome biology. Additionally, it is likely that tetragnathid spiders will have many novel genes and gene families, as this family has not previously been the subject of deep sequencing efforts.

Interestingly, this study shows that the majority of proteins identified in the *T. versicolor* venom proteome are sexually dimorphic (~87 %), with 4,908 distinct proteome entities only in adult males (Fig. 1). Most of these had no detected corresponding transcripts in female venom gland tissues and are not present in female venom cocktails. However, a small number of sequence reads from females map to transcripts of male-only peptides (and vice versa), indicating these may be expressed in females but not translated or not incorporated into the female venom cocktail. Of the "high molecular weight", male-only proteins in the venom proteome, only three of 23 corresponding transcripts are not differentially expressed between the sexes, as indicated by non-significant GFOLD values. While the males have more unique components, there are 4,269 unique female proteome entities, including small number of unique low molecular weight peptides.

The unique peptides in female venoms are homologous to other spider venom peptides, range in size from 5.9 to 7.9 kDa, and have ICK motifs (-C6C-CC-C-C-) that are consistent with them functioning as neurotoxins involved in prey immobilization. The biased presence of potentially toxic peptides in female venoms is consistent with observations of differences between males and females in concentration of low molecular weight components (*Binford, Gillespie & Maddison, 2016*, Fig. 1C). This pattern mirrors within sex, among species differences in Hawaiian *Tetragnatha* that have evolved differences in feeding biology. Specifically, as part of an adaptive radiation within Hawaiian *Tetragnatha,* a clade lost web-building behavior and evolved to be wandering foragers with an associated shift in dietary niche. With that evolutionary transition to wandering the lineage underwent a coincident reduction in low molecular weight venom peptides (*Binford, 2001*). Given that evolutionary shifts in low molecular weight peptides in venoms appear to occur in association with shifts in feeding biology, the lack of detection of venom peptides in males may be best explained by differences in adult niche that lead to a reduction in male reliance on venom peptides for prey immobilization (*Binford, Gillespie & Maddison, 2016*).

The more striking dimorphism that is less easily explainable by differences in dietary niche is in the male specific novel proteins, the "cluster 6" proteins. These belong to a single gene family with estimated molecular weights corresponding to proteins detected with 1-D protein gels across a comparative sampling of *Tetragnatha* (Fig. 1, *Binford, Gillespie & Maddison, 2016*). The rationale for suggesting a possible role beyond feeding is that these "cluster 6" components comprise a high proportion of the male-specific proteins (12 of 23 unique proteins; next largest family 2 of 23; Table 1A), suggesting an important functional role unique to males. While BLAST searches of these male-specific proteins did not detect homology to known sequences, predicted structural homology to Argos proteins that bind ligands to epidermal growth factors suggests potential for binding to small proteins. Moreover, functional annotations and GO enrichment analyses suggest hormone-related functions. Due to the degree of similarity in motifs, and likely homology, between Argos

and the novel male-only "cluster 6" proteins discovered in *T. versicolor*, we propose the name *Argoinonui* (*Argo* for the Argos protein, *ino* is Hawaiian for "venom", and *nui* is Hawaiian for "large") for this high-molecular weight venom gene family.

The high molecular weight components in male *T. versicolor* venom may be pervasive in the genus *Tetragnatha*, based on previous 1-D gel studies of venom peptide diversity (*Binford, Gillespie & Maddison, 2016*). One striking aspect of this genus of spiders is their very unusual sexual behavior: While courtship in most spiders involves an elaborate and extended period of vibrational or visual communication, in most *Tetragnatha* there is little evidence for communication prior to the male and female approaching each other. They connect physically by spreading the chelicerae wide and locking fangs (Fig. 1A), involving a dorsal spur on the male chelicerae. The cheliceral-locking mechanism apparently precludes the need for epigynal coupling and is associated with secondary loss of a sclerotized epigynum (*Levi, 1981*). These alterations of mating morphologies combined with the presence of male-only components of the venom cocktail that are likely not used in feeding or defense, lead to a hypothesis that the unique male components in venom play some as yet undescribed role in mating biology (*Binford, Gillespie & Maddison, 2016*).

While sexual differences in venom composition driven by adult niche are likely, venoms are also known to play a role in sexual biology (*Polis, 1990*) and thus may be under the influence of sexual selection. As secreted molecules with intra-individual functionality, they have potential for biological roles in sex (*Binford, Gillespie & Maddison, 2016*). The origin of the male-specific proteins appears to have been coincidental with the origin of the unusual premating cheliceral-locking behavior by which these spiders intertwine their fangs while mating (Fig. 1A). Members of the family Tetragnathidae are secondarily haplogyne, having lost much of the complexity in male and female genitalia that often functions in maintaining species boundaries and mate recognition, evolving via sexual selection by female choice. The presence of cheliceral locking during mating in many tetragnathid species provides an alternative mechanism upon which sexual selection and mate recognition may have evolved. This has been demonstrated in the tetragnathid species *Leucauge venusta* where tactile stimulation of females is accomplished via specialized male setation during cheliceral locking (*Aisenberg, Barrantes & Eberhard, 2015*).

Two pieces of evidence suggest potential sexual roles for the high molecular weight venom components. First, the possibility that the two groups of high molecular weight proteins are not involved in feeding or defense is supported by the significant homology of these components to hormone processing peptidases (BLAST *e*-value ~0.0). If true, then we would expect to find these high molecular weight proteins in all spiders that show cheliceral-locking behavior, but not in those without; while preliminary data for a small number of Hawaiian and mainland *Tetragnatha* support this hypothesis (*Binford, Gillespie & Maddison, 2016*), clearly more data are needed. Second, if the high molecular weight components in the venom are playing a sexual function, then we might expect that recognition could be compromised at some level in taxa that display cheliceral-locking behavior but do not have high molecular weight proteins. Here again, an intriguing observation in support of this argument is the finding of prolonged male-to-male cheliceral locking and mating attempts in the tetragnathid spider *Doryonychus raptor* (*Gillespie, 1991*),

a species which lacks dimorphic venom(*Binford, Gillespie & Maddison, 2016*), suggesting that recognition is less complete than in most other taxa that employ this mating strategy and exhibit dimorphic venoms.

While it is interesting that the cheliceral-locking mating behavior and dimporphic venoms coincide, especially considering mate recognition is only known to be complete in species possessing both, several caveats exist. The male-only peptides present in *T. versicolor* are large (>100 kDa), unlike known volatile pheromones (*Wyatt, 2014a*), and a mechanism for their transfer to females is presently unknown. Additionally, a previous study of spider venom sexual dimorphisms in the distantly related and non-cheliceral locking species *Phoneutria nigriventer* (Araneae: Ctenidae) indicated male-only, high molecular weight components (*Herzig, Ward & Santos dos, 2002*), a pattern similar to *T. versicolor*. No known mating mechanism has been ascribed to the *P. nigriventer* dimorphism. Much like Hawaiian *Tetragnatha*, both sexes of *P. nigriventer* wander in search of prey, but we do not know whether these genes are homologous and cannot speculate on any shared function.

Regardless of the functions, a striking sexual dimorphism is present in all species of *Tetragnatha* examined to date. Our results provide the first sequence-level investigation of the family Tetragnathidae and highlight the diversity of spider-produced chemicals awaiting discovery in understudied groups.

## CONCLUSIONS

The genus *Tetragnatha* exhibits striking sexual dimorphisms in venom composition (*Binford, Gillespie & Maddison, 2016*). Herein, we have documented the specifics of this dimorphism in the species *T. versicolor* using deep sequencing of transcriptomes and mass spectrometric proteomic validation of transcript translation. While the results are still preliminary, the venom of *Tetragnatha* spiders may function in both mate recognition as well as adaptive specialization for prey, in which case venom could provide insights into the genomic underpinnings of adaptive radiation as well as the interplay between plasticity and variability in fostering species proliferation. The venom of *T. versicolor* includes proteins and peptides comprising a wide array of molecular weights, as well as many novel compounds. Many of the dimorphic components cannot be associated with any currently characterized peptides. Previous work has demonstrated sexually dimorphic venoms can facilitate each mature sex occupying different feeding niches, and several of the female-only components present in *T. versicolor* are homologous to traditional feeding and defensive toxins. Males possess several unique gene families, many of which cannot be annotated. A newly-discovered high molecular weight and male-only gene family, deemed *Argoinonui*, is associated with gene ontologies relating to hormone processing and regulation and has FFAS indicated homology to the *Drosophila* protein *Argos*, an epidermal growth factor associated protein.

## ACKNOWLEDGEMENTS

The authors would like to thank David and Diana Binford for help collecting on the farm; Linda Breci and Cynthia David for proteomic work at the RDI Analytical/Biological Mass

Spectrometry Core, University of Arizona; and Chase Magsig, Aayushi Patel, and Augustus Floyd for help collecting North Carolina Spiders.

### Funding

This work was funded by East Carolina University; the University of California, Berkeley; and Lewis and Clark College. The funders had no role in study design, data collection and analysis, decision to publish, or preparation of the manuscript.

### Grant Disclosures

The following grant information was disclosed by the authors:
East Carolina University.
University of California, Berkeley.
Lewis and Clark College.

### Competing Interests

The authors declare there are no competing interests.

### Author Contributions

- Pamela A. Zobel-Thropp conceived and designed the experiments, performed the experiments, analyzed the data, prepared figures and/or tables, authored or reviewed drafts of the paper, approved the final draft.
- Emily A. Bulger performed the experiments, analyzed the data, authored or reviewed drafts of the paper, approved the final draft.
- Matthew H.J. Cordes performed the experiments, analyzed the data, contributed reagents/materials/analysis tools, prepared figures and/or tables, authored or reviewed drafts of the paper, approved the final draft.
- Greta J. Binford conceived and designed the experiments, performed the experiments, contributed reagents/materials/analysis tools, authored or reviewed drafts of the paper, approved the final draft.
- Rosemary G. Gillespie conceived and designed the experiments, contributed reagents/materials/analysis tools, authored or reviewed drafts of the paper, approved the final draft.
- Michael S. Brewer conceived and designed the experiments, performed the experiments, analyzed the data, contributed reagents/materials/analysis tools, prepared figures and/or tables, authored or reviewed drafts of the paper, approved the final draft.

### DNA Deposition

The following information was supplied regarding the deposition of DNA sequences:
 The raw sequencing data have been deposited in the NCBI Short Read Archive (SRA accession number SRP118124).

## Data Availability

Brewer, Michael (2017): Publication Report for EB_215046_DB052615.xls. figshare. Dataset. https://doi.org/10.6084/m9.figshare.5378308.v1.

Brewer, Michael (2017): Protein Report for EB_215046_DB052615.xls. figshare. Dataset. https://doi.org/10.6084/m9.figshare.5378299.v1.

Brewer, Michael (2017): TversicolorDimorphism_GFclustering.tar.gz. figshare. Dataset. https://doi.org/10.6084/m9.figshare.5420986.v1.

Brewer, Michael (2017): TVBF_both_Trinotate_annotation_report_withUniref90.xls. figshare. Dataset. https://doi.org/10.6084/m9.figshare.5421655.v1.

Brewer, Michael (2017): TversicolorDimorphism_GFOLD.tar.gz. figshare. Dataset. https://doi.org/10.6084/m9.figshare.5421517.v1.

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
