# Peer review of "Sexually dimorphic venom proteins in long-jawed orb-weaving spiders (Tetragnatha) comprise novel gene families"

_PeerJ, doi:10.7717/peerj.4691_

## Round 0.1 · original submission · Major Revisions

Overall I really enjoyed reading this manuscript and agree with both reviewers that this is interesting data on a timely topic. I found the manuscript to be very well written and nicely set out. The main area that I think needs improvement (suggested by both reviewers) is that support for the role of these large proteins in sexual communication is rather weak. This detracts from the discussion, which will need to be restructured to present alternative hypotheses about what these male specific proteins may be doing. Please see further comments around ideas for the discussion from both reviewers. Finally, please take into account the other minor comments from both reviewers, who did a great job of reviewing this manuscript.

Reviewer 1 ·

Basic reporting

The following statement that wandering behaviour reduces dimorphism in feeding biology requires citation or additional evidence, as it is central to the focus of the paper:
“… both males and females wander in search of prey, thus reducing dimorphism in feeding biology”.

2. The reference {Maretic:1964um} (Line 48) is not formatted.

Experimental design

no comment

Validity of the findings

1. The results do not fully support the hypothesis proposed – that is, that the unique male components in venom play some as yet undescribed role in mating biology. The null hypothesis – that the unique male components in venom do not play any role in mating biology, or perhaps play a role in male-specific predator/prey interactions (as is proposed for the female specific venom components), based on the data presented, cannot be discounted. The GO annotation results, loosely linking the proteins to “hormone function” is tenuous evidence at best and has been overinterpreted to fit the hypothesis.

The inability of the presented data to discount the null hypothesis or alternative hypotheses appears to be acknowledged in some part by the authors. However, the idea that the unique male components in venom play some as yet undescribed role in mating biology still remains the central theme around which the paper is written. The data clearly showing sexual dimorphism in venom composition, by itself is of considerable interest, and if this were the central theme of the paper (with the idea that the unique male components in venom play some as yet undescribed role in mating biology confined to the discussion and clearly indicated as speculative), I would have, without hesitation, recommended the articles acceptance in PeerJ.

I suggest the article be restructured with these points in mind. i.e. to follow the layout in the current abstract (which is well written and well-structured for the data). In the introduction deemphasise the idea of sexual selection - much of this could be shifted to the discussion and more clearly indicated as speculation.
Much of the Discussion and Conclusions section could simply be removed or reworded e.g. Lines 335, 408, 445 - the words "support" and "supporting" are used. In this context, it more accurate to say "consistent with". At the moment these sections do not clearly show that the points made are essentially speculative.

2. A final related point that I suggest needs to be covered in the Discussion: A high molecular wight protein seems an unlikely candidate as a pheromone… Small volatile compounds can act as pheromones while proteins typically take effect only on injection. Is there any evidence for injection of venom on mating?

Additional comments

While my review may come across as overly negative, I do think the article is exceptionally well written. However I am concerned that the data has been overinterpreted to fit a particular hypothesis. I have suggested a "major revision", but only in terms of restructuring what is already presented in the article.

Reviewer 2 ·

Basic reporting

ok

Experimental design

ok

Validity of the findings

the data are robust, but the conclusions should be improved. Please see my suggestions under general comments for the author.

Additional comments

The manuscript entitled "Sexually dimorphic venom proteins in long-jawed orb-weaving spiders (Tetragnatha) with potential roles in sexual interactions" submitted by Zobel-Thropp et al. is a very interesting study that investigates intraspecific variations in the venom composition based on the gender of the spiders. The authors use transcriptomic and proteomic techniques to identify the differences in venom composition between male and female Tetragnatha species. While the methods that the authors employed are state of the art and the experimental part and data analysis are carried out appropriately, I am not quite convinced by their discussion. In particular, the suggestion that the large molecular weight components that were specific to male venom, would be involved in sexual communication sounds rather unlikely to me. In arthropods, molecules used for chemical communication (incl. sexual pheromones) are usually small and volatile, so they can easily reach the respective recipient. In the present case, I don't quite see how the > 100 kDa male-specific components could be used for chemical communication. I would strongly doubt that such large proteins are volatile. And, while the authors mention the cheliceral locking during mating, according to my understanding that locking does not involve the injection of venom components from one individual to another. Thus, how do the authors think that these proteins are "communicated" or transmitted between the spiders? Or by what other mechanism could they be involved in the chemical communication? I think the authors need to either support their hypothesis more strongly or get rid off it entirely.

What is also quite interesting to me is that this study is not the first one to show some large proteins in the venom of male but not female spiders, as this has already been reported for Phoneutria nigriventer (see Herzig et al., Toxicon 2002). However, in the latter case, there is no cheliceral locking involved in the mating of P. nigriventer. I would therefore strongly question whether the large male-specific components are related to the cheliceral locking mechanism. The interesting question would be what else could they be doing? So the authors should discuss this in more detail.

I also have some minor issues that need to be fixed.

• L. 60: no bracket in front of Kuhn-Nentwig
• L.100: "evidence for" instead of "evident"
• L.134 ff: It would be interesting if the authors could include the venom yields for the different sexes.
• L.266: the authors should provide the appropriate references for NCBI and ArachnoServer
• L.435: The statement " we have shown that the venom of Tetragnatha spiders can potentially serve in in both mate recognition as well as adaptive specialization for prey" has to be toned down, since this has not been shown by the authors - it's only a hypothesis (which I am not quite convinced of).
• L. 455 ff. In the references section, the authors need to make sure that ALL of the genus and species names are in italics. Furthermore, for some of the references, they have all words in the title start with capital letters, but not so for other references. This needs to be consistent for all references.
• L.590: the current version of WSC is 18.5, so this needs to be updated.

---

## Round 0.2 · Minor Revisions

This revision of your manuscript was a pleasure to read and benefitted from the changes you made from the previous version. From careful reading I have no major comments on the manuscript, but some of the references need a little formatting. After you tidy up a few of the other minor changes suggested by the reviewer it should be acceptable for publication.

Reviewer 2 ·

Basic reporting

no comment

Experimental design

no comment

Validity of the findings

no comment

Additional comments

The revised manuscript entitled "Sexually dimorphic venom proteins in long-jawed orb-weaving spiders (Tetragnatha) comprise novel gene families " submitted by Zobel-Thropp et al. has now been substantially improved in comparison with the previous version.

Except for some minor points (see below), I think the manuscript is almost ready for publication.

• L75-76: Another explanation for the "higher mammalian neurotoxic" activity could just be that males are a lot smaller and therefore yield substantially smaller venom amounts. I don't think that the chosen reference (Maretić, Levi & Levi, 1964) provides sufficient evidence for either case, as they don't really compare similar amounts of venoms from both sexes. Hence, other references need to be included to support this statement or it has to be removed.
• L92: "that are not (found/present ?) in females"
• L.360ff: "The high molecular weight components in male T. versicolor venom may be characteristic of the genus Tetragnatha" - based on similar observations from P. nigriventer venom, that statement is not true. Or did the authors mean "characteristic for male T. versicolor within the genus Tetragnatha"?
• L430: should that read "toxins" instead of "venoms"?
• Re. my previous comment on the venom yields: in case the authors have recorded the average male and female venom yields (rather than the individual yields), such data would also be interesting to report.
• Re. the terminology of peptides/proteins, I think the common rule is that anything above 10 kDa is termed protein, whereas anything below is termed a peptide. Hence, the high-molecular weight male venom compounds should be termed as proteins (rather than polypeptides).
• L519: should the "11" be in front of "Edited" or is this a typo?
• L486: some references still have all words starting with capital letters in the title, which should be changed.

---

## Round 0.3 · accepted · Accept

The current version of the manuscript has made all changes that were suggested from the last minor revision decision. Overall, the readability of the manuscript is excellent and I feel the manuscript will make a nice contribution to ecology and evolution. I understand that you that you had to pool venom and could not record individual amounts for each male. I feel that the manuscript is now acceptable for publication.

#